# Onion Peel Waste Mediated-Green Synthesis of Zinc Oxide Nanoparticles and Their Phytotoxicity on Mung Bean and Wheat Plant Growth

**DOI:** 10.3390/ma15072393

**Published:** 2022-03-24

**Authors:** Shreya Modi, Virendra Kumar Yadav, Nisha Choudhary, Abdullah M. Alswieleh, Anish Kumar Sharma, Abhishek Kumar Bhardwaj, Samreen Heena Khan, Krishna Kumar Yadav, Ji-Kwang Cheon, Byong-Hun Jeon

**Affiliations:** 1Department of Microbiology, Shri Sarvajanik Science College, Mehsana 384001, India; shreyamodi20@gmail.com; 2Department of Microbiology, School of Sciences, P P Savani University, Surat 394125, India; yadava94@gmail.com; 3Department of Environmental Sciences, School of Sciences, P P Savani University, Surat 394125, India; nishanaseer03@gmail.com; 4Department of Chemistry, College of Science, King Saud University, Riyadh 11451, Saudi Arabia; aswieleh@ksu.edu.sa; 5Department of Biotechnology, School of Sciences, P P Savani University, Surat 394125, India; anish.sharma@ppsu.ac.in; 6Department of Environmental Science, Amity School of Life Sciences, Amity University, Gwalior 474001, India; bhardwajak87@gmail.com; 7Research and Development Centre, YNC Envis Pvt. Ltd., New Delhi 110059, India; samreen.heena.khan@gmail.com; 8Faculty of Science and Technology, Madhyanchal Professional University, Bhopal 462044, India; envirokrishna@gmail.com; 9Department of Earth Resources & Environmental Engineering, Hanyang University, 222-Wangsimni-ro, Seongdong-gu, Seoul 04763, Korea; jkcheon@hanyang.ac.kr

**Keywords:** phytotoxicity, phytochemicals, onion peel extract, *Vigna radiate*, *Triticum aestivum*

## Abstract

Nanoparticles and nanomaterials have gained a huge amount of attention in the last decade due to their unique and remarkable properties. Metallic nanoparticles like zinc oxide nanoparticles (ZnONPs) have been used very widely as plant nutrients and in wastewater treatment. Here, ZnONPs were synthesized by using onion peel and characterized by various sophisticated instruments like Fourier transform infrared spectroscopy (FTIR), dynamic light scattering (DLS), and field emission scanning electron microscopes (FESEM). FTIR confirmed ZnONPs synthesis due to the formation of the band in the region of 400–800 cm^−1^, while FESEM confirmed the spherical shape of the particles whose size varies in the range of 20–80 nm. FTIR revealed several bands from 1000–1800 cm^−1^ which indicates the capping by the organic molecules on the ZnONPs, which came from onion peel. It also has carbonyl and hydroxyl groups, due to the organic molecules present in the *Allium cepa* peel waste. The average hydrodynamic size of ZnONPs was 500 nm as confirmed by DLS. The synthesized ZnONPs were then used as a plant nutrient where their effect was evaluated on the growth of *Vigna radiate* (mung bean) and *Triticum aestivum* (wheat seeds). The results revealed that the germination and seedling of mung and wheat seeds with ZnONPs were grown better than the control seed. However, seeds of mung and wheat with ZnONPs at median concentration exposure showed an enhancement in percent germination, root, and shoot length in comparison to control. Thus, the effect of ZnONPs has been proved as a nano-based nutrient source for agricultural purposes.

## 1. Introduction

Nanoparticles have gained a huge amount of attention in the last decade due to their high demand in the fields of environmental cleanup, micronutrients for plants and biofertilizers, and electronics [1,2]. The increasing use of nanoparticles and their occurrence in the environment has made it imperative to elucidate their impact on the environment [3,4]. With the progression of nanotechnology, the number of consumers has increased very dramatically for the use of different nanoparticles in agriculture. Due to the broad-scale use of nanoparticles, these nanomaterials can be expected to be present in the environment, raising concerns about their effects on plants and animals. Zinc oxide nanoparticles (ZnONPs) have a specific surface area, high pore volume, low toxicity, and long life-span, and thus are being used as a promising nanomaterial as antibacterial, cosmetics additive, chemical absorbents, catalysts, and polymer additives [5,6]. Zinc is an essential micronutrient that is absorbed in the form of divalent cations. It is required for the growth of the plant, protein synthesis, maintenance of membrane integrity, and energy production [7]. Zinc is also required for the activation of various enzymes which are used for the formation of chlorophyll as well as auxin synthesis. In addition to this, it also has a role in the activation of enzyme-like dehydrogenase, phosphoryl hydrolase, peptide, and proteases [8]. Zinc nanoparticles alone or in the form of oxides are easily adsorbed by the plant roots due to their small size and become easily available to every part of the plant. Several other metallic nanoparticles have also been used for the same purposes like magnesium nanoparticles, Cr_2_O_3_ nanoparticles by Ghotekar et al., and CoFe_2_O_4_ NPs for wastewater treatment, etc. [9,10,11,12,13,14,15].

Soil properties restrict the uptake of nutrients from the NPs by plants due to variation in dissolution, aggregation, and change in surface properties of NPs in soil [16]. The chemical-mediated synthesis of ZnONPs employs harsh chemicals, which have adverse effects on the environment. Thus, biological synthesis is one of the most eco-friendly methods for the synthesis of ZnONPs. Moreover, if the synthesis is mediated by using waste materials like vegetable waste, then the synthesis process not only becomes green but also becomes environmentally-friendly. One such vegetable waste is onion, i.e., onion waste, which is commonly generated in every house, especially in a country like India. Onion is considered to have numerous phytochemicals, flavonoids, and enzymes that help in the synthesis of nanoparticles [17,18,19]. Alamdari et al., 2020 reported the synthesis and characterization of ZnONPs by using leaf extract of *Sambucus ebulus.* These particles were wurtzite nanocrystal structures, whose average size was 17 nm. It was applied further as an antimicrobial agent and dye removal from wastewater [20]. Dogan and Kocabaş reported the green synthesis of ZnO nanoparticles with *Veronica multifida* and performed their antimicrobial activities [21]. In addition, Modi and Fulekar. (2020) reported the synthesis of ZnO NPs by using *Allium sativum* skin extract. Besides this, there are several pieces of literature where ZnO NPs effect was observed on the growth of various crop and non-crop plants [22,23]. In the absence of Zn or under deficiency, the plant may exhibit adverse effects, like chlorosis, stunted growth, etc. Nemcek et al., 2020, Faizan et al., 2020, Plaksenkova et al., 2020, and Rajput et al., 2018, reported the effect of ZnONPs and bulk on the various crop and non-crop plants. From all the above work, it was found that most of the synthesis was done by using plant extract; only a few authors have reported the synthesis of ZnONPs by using waste like an onion peel, while all of them have used it for antimicrobial activity [24,25,26,27].

Here in the current research work, authors have synthesized ZnONPs by using onion peel waste and observed its effect as a micronutrient on the growth of the Mung bean and wheat plant. The synthesized nanoparticles were characterized by the various sophisticated instruments for revealing the detailed characteristics of ZnONPs to confirm the nano-size nature of particles. The synthesis of the ZnONPs was biological and a green route, which makes them biocompatible. In the present study, plant mediated synthesized ZnONPs were used to check their toxicity and impact on the seed germination and root length of the Mung bean (*Vigna radiate* L.) and Wheat (*Triticum aestivum*) seeds using standard methods.

## 2. Materials and Methods

### 2.1. Materials

Zinc chloride (Sigma Aldrich, Darmstadt, **Germany****),** sodium hydroxide (SRL, Gujarat, India), onion skin from the vegetable market, Gandhinagar, Gujarat, Mung bean and Wheat seeds.

### 2.2. Synthesis of ZnO Nanoparticles by Using Onion Peel Extract

#### 2.2.1. Preparation of Onion Skin Extract

The onion peel waste collected from the vegetable market was washed several times with double distilled water in order to eliminate the dust particles associated with it. Furthermore, after washing, it was dried in an oven at 40 °C for 7 days. Once it was completely dried, it was converted into powder by using a home grinder (Jaipan with model no JP_KKMG 750-Watt). Furthermore, it was sieved in order to eliminate any larger particles that failed to grind completely during the previous step. Furthermore, about 10 g of dried onion peel powder was weighed and soaked in the DDW, in a 250 mL Erlenmeyer flask. In addition, about 100 mL of 80% ethanol was added into the flask. In the next step, the above mixture was heated at 60 °C for 20 min along with continuous stirring for 24 h. Afterwards, the completion of stirring time, the mixture was cooled to room temperature (RT), followed being bypassed through the Whatman filter paper No. 1. About 50 mL of filtrate was taken for the synthesis of NPs, while the rest of the filtrate was evaporated in a rotary evaporator at 45 °C until the extracts get completely dried. After evaporation, the extract was stored at 4 °C for future application. The qualitative and quantitative analysis of onion skin extract was done using standard methods [28].

#### 2.2.2. Synthesis of ZnONPs

About 200 mL of 2 mM ZnCl_2_ solution was prepared and kept in a stirrer for 20 h. The parameters were optimized and the best pH value was observed at pH 8 for the synthesis applications, so, throughout the experiment, the pH of the solution was maintained at 8 by using 1M NaOH solution. About 30 mL of onion peel extract solution were added drop wise to the above solution under constant stirring conditions and the color of the reaction mixture was changed after 30 min of incubation time. The solution was left in stirring conditions for 4 h after the incubation time confirmed the synthesis of ZnONPs. The mixture was centrifuged at 7000 rpm for 5 min, in order to separate the solution from the residue. The solution was decanted while the precipitate was retained. The precipitate was washed several times with DDW and ethanol in order to eliminate any impurities associated with their surface. The obtained precipitate was dried in an oven at 80 °C till complete dryness. Finally, the obtained powder was calcined at 400 °C in a muffle furnace by slowly increasing the temperature to 5 °C per min.

#### 2.2.3. Characterization of ZnONPs

The synthesized ZnONPs were analyzed by the electron microscopy field emission scanning electron microscope (FESEM) and Transmission electron microscope (TEM) for their morphology properties. The particle was further analyzed by the Fourier-infrared spectroscopy (FTIR), and particle size distribution (PSD) was analyzed by the dynamic light scattering (DLS). All these analyses helped in the identification and the confirmation of formation of ZnONPs. For FESEM analysis, dried powder samples were placed on the double-sided carbon tape, which in turn was placed on the aluminum stub. The analysis was done by NOVA NANOSEM (FEI, USA). EDX analysis was also done to characterize the surface of nanoparticles. The PSD of ZnONPs was done by (Model: NPA152-31A-0000-000-90M, make: Metrohm, Herisau, Switzerland) by sonicating a pinch of sample in the double-distilled water for 10 min in an ultrasonicator (Sonar, 40 kHz). For FTIR analysis, about 2 mg of ZnONPs was mixed with 198 mg of KBr, and a solid pellet was prepared by using a mechanical press. The measurement of the sample was done in the range of 400–4000 cm^−1^ at a resolution of 1 nm by using Model: SP-65, Make: Perkin Elmer While TEM analysis was done to find the size of the ZnONPs by a drop-casting technique, the dispersed sample from DLS was dropped with a micropipette on the carbon-coated copper grids. The morphological measurement of the ZnONPs was carried out by FEI Techni F20 G2, (USA).

#### 2.2.4. Phytotoxicity of the ZnONPs on the Plant Growth 

In this experiment, onion skin extract mediated synthesized ZnONPs were used. The solutions having different concentrations (100, 200, 300, 400, 500, 600, 700, 800, 900, 1000, 1200 ppm) of ZnONPs were prepared. The mung bean and wheat seeds were procured from the market. The seeds were cleaned properly and kept in a dry place in the dark under the room temperature before use. 

#### 2.2.5. Seedling Exposure

The seeds were checked for their viability by suspending them in double-distilled water. The seeds that are settled to the bottom were selected for further experimental study. The seeds were rinsed in double distilled water thrice. The seeds were sonicated for 2 h in prepared nanoparticles suspensions (100, 200, 300, 400, 500, 600, 700, 800, 900, 1000, 1200 ppm) using a sonicator (Sonar, 40 kHz) and kept in the same solution for 24 h [29]. The next day, the soaked seeds were put in prepared pots and further growth studies were checked out. 

#### 2.2.6. Physico-Chemical Properties of Soil

The soil used in the experiment was collected from the agricultural field located in sector 30, Gandhinagar, Gujarat. The texture of the soil is loamy sand, which was further sieved, and the sample was taken for further physico-chemical characterization. The results of physico-chemical analysis are shown below in Table 1.

#### 2.2.7. Root and Shoot Length

Root length was taken from the point below the hypocotyls to the end of the tip of the root. Shoot length was measured from the base of the cotyledons. A thread and scale were used to measure the root and shoot length.

#### 2.2.8. Seed Germination Test

The seed germination rate (RSG) and relative root growth (RRG) was calculated using the equation and germination index (GI).
Relative seed germination Rate = (Sc/Ss) × 100
Relative root growth = (Rs/Rc) × 100
Germination Index = (RSG/RRG) × 100
where Ss is the number of seeds germinated in sample, Sc is the number of seeds germinated in the control, Rs is the average root length in the sample and Rc is the average root length in the control. 

#### 2.2.9. Fresh and Dry Weight

The fresh weight of root and shoot of seedlings was determined by weighing of the roots and shoots separately on electric balance. After taking the fresh weight, the seedlings were kept in an air oven at 60 °C for 48 h; then, the weight of dry matter was recorded. 

## 3. Results and Discussion

### 3.1. Morphological Analysis of Synthesized ZnONPs by FESEM and TEM

The synthesized ZnONPs are spherical shaped whose size is varying from 40–120 nm, as shown in Figure 1. All the particles are spherical and smaller in size, but they have aggregated to form a large structure of size above 100 nm, while some of the nanoparticles got fused due to calcination at high temperature. Aggregated spherical structures of synthesized ZnONPs were also reported by Talodthaisong et al., where the synthesized ZnONPs were decorated with Gamma-Aminobutyric Acid, derivatives of curcumin and AgNPs [30]. 

The particle size of synthesized ZnONPs varied from 76.40 ± 16.31 nm, while their morphology was spherical along with a few rod like structures [31]. Haque et al. 2020 reported ZnONPs synthesis from neem leaf extract of size 25–34 nm. The small size was obtained due to the presence of terpenoids present in the neem leaf extract [32]. 

A typical TEM image is shown in Figure 2, which is spherical to triangular in shape. Figure 2 is showing a TEM image of ZnONPs at 0.1 um, the particle size is varying from 20–80 nm while a few of them are showing aggregation due to the very small size of the particles. The morphology was also supported by the FESEM image of the particle. Similar results were also reported by Modi and Fulekar [28]. Haque et al. reported ZnONPs of synthesis from neem leaf extract of hexagonal, spherical and rod-like, along with agglomeration to form large particles [32]. Jan et al., [33] also reported the synthesis of ZnONPs by using aqueous leaf extract of *Aquilegia pubiflora*. Here, the average size of the particles was 34.23 nm, which was later on assessed for their antiproliferative activity against HepG_2_ Cells, inducing reactive oxygen species. Similarly, Iqbal et al. and Murali et al. reported the plant based biological synthesis of ZnONPs and reported their multiple applications. For instance, the former group reported the synthesis of ZnONPs by *Elaeagnus angustifolia* L. leaf extracts where the phytochemicals are assumed to serve as a non-toxic source of reducing and stabilizing agents [34,35].

### 3.2. Particle Size Distribution of the ZnONPs

A typical particle size distribution graph of ZnONPs is shown in Figure 3. The particles size distribution of the synthesized ZnONPs was varying from 100 nm to 1200 nm. This was so because smaller particles were aggregated and formed large particles. The average particle size was 450 nm in diameter and the zeta potential was 16.59 mv, while the polarity was found to be positive for the onion synthesized ZnONPs. The results were in agreement with previously synthesized nanoparticles by Modi and Fulekar 2020 [28]. Aldalbahi et al., 2020 reported the ZnONPs by green route, where the average particle size was 94.36 nm (in diameter) and the major population (93.3% based on the intensity) of the particle was in the range of 101.4 nm (in diameter) nm while the minor population (6.7% based on the intensity)of the particle was 4969 nm (in diameter) [36]. Haque et al. 2020 also synthesized the ZnONPs, by using *Azadirachta indica* (neem) leaves, and obtained similar results [32].

The formation of larger size particle or nanoaggregates is due to the formation of hydrodynamic layers around the particle in liquid medium. Because of this, the overall particle size appears to be larger than the actual size of nanoparticle. It may be also due to the presence of some dust particle.

### 3.3. FTIR Study of Synthesized ZnONPs

Figure 4 shows the typical diagram of FTIR of ZnONPs synthesized by onion peel waste. The peaks at 510.3, 603.9 and 790.1 cm^−1^ are attributed to the Zn-O bond weak and strong stretching and indicate the formation of the particles. The band at 1122.5 cm^−1^ is due to the presence of an amide group in the nanoparticles from the peel, while a sharp and deep band at 1625.7 cm^−1^ is due to the C = O stretching and due to the presence of a water molecule (−OH) in the nanoparticle. A small peak near 2500 cm^−1^ is attributed to the adsorption of atmospheric carbon dioxide on the sample. A broadband at 3420.1 cm^−1^ indicates the presence of hydroxyl molecules in the synthesized ZnONPs. Similar results were also obtained by Modi and Fulekar [28]. Rajendran et al. and Belely et al. have also reported similar results for the synthesis of ZnONPs obtained by a chemical method and a green route, respectively [37,38].

### 3.4. Physicochemical Analysis of the Soil 

The physico-chemical study was done for the soil used during the experiment. The obtained results are represented in Table 1. The percentage of seed germination was significantly affected by the interaction of ZnONPs. In comparison with the control and ZnONPs treated seeds, there was increased germination. Table 2 and Table 3 show the results for Mung bean and wheat, respectively.

The total nitrogen content in the soil was 0.87%, which is generally considered high for soil, while its phosphorus content was 5, which is considered low. The pH of the soil was 7.6 i.e., normal or near to neutral, while conductivity was 1.35, which indicates the presence of high salt content. Sulphur (S), zinc (Zn), and iron (Fe) were present in the medium range whose concentration was 16.5 ppm, 4.20 ppm, and 8.4 ppm, respectively. The concentration of Mn and Cu was 29.1 and 1.92 ppm in the soil, which is generally considered high.

The effect of ZnONPs dosage was observed on the growth of the Mung bean plant. There was one control plant also against which the experimental plant was analyzed. The germination of seedling was 90% in the control, while it was 100% from 50–600 ppm of ZnO NPs concentration. When the ppm of ZnONPs reached 800, 1000 and 1200, the seed germination again reduced to 90%, while, at 1200 ppm, the growth reduced to 80%. This indicates that, in the absence of ZnONPs, growth is reduced but required Zn at desired ppm for the optimum growth. While the higher concentration leads to the toxic effect on the plant, due to which there is a decrease in the seed germination, which is reduced to 80%. This indicates that the high dose of Zn is toxic for the growth of the plant as the Zn itself is a heavy metal. The maximum root length of the sapling was seen with the 100 ppm of ZnONPs, which was 9.49 cm, while the maximum shoot length was obtained when the ZnO NPs concentration was 200 ppm, which reached up to 15.88 cm. The maximum fresh weight of root was obtained at 50 ppm, which was 0.108 gram, and the lowest was 0.056 gram in control sapling. While in the case of shoots, the highest fresh weight of shoot was obtained at 150 ppm, which was 1.734 and lowest in control 1.182 grams. The maximum dry weight in the case of mung bean root was 0.042 at 150 ppm and lowest i.e., 0.019 gram at 1200 ppm obtained with ZnONPs. However, in the case of the dry weight of shoots, the highest was at 600 ppm i.e., 0.215 grams while the lowest was in the case of control i.e., 0 ppm, which was 0.114 grams in the case of a mung bean plant. Savassa et al., Szollosi et al. and Hajra et al. reported the effect of ZnONPs on the growth of the mung bean plant [39,40,41]. They also reported that Zn is a micronutrient required by the plant and promotes the growth of the mung bean plant, but a higher concentration of ZnONPs leads to phytotoxicity, due to which the effect on the growth of the plant is observed either in the form of short root, short soot length, their wet weight and dry weight.

In the case of wheat seed germination in control, growth was just 50% while the seed germination increased with a gradual increase in the concentration of the ZnONPs. At 50 and 100 ppm, the seed germination was 70%, while, at 150 and 200 ppm, the growth was 80% while the growth reached 100% at 400, 600 and 800 ppm. After that, the seed germination was decreased, i.e., at 1000 ppm, the growth was just 70%, while, at 1200 ppm, it was only 70%. Initially, the seed germination was less when low Zn was there, and slightly increased with an increase in the ppm. Once the ppm was high, it became toxic to the plant as it acts as a heavy metal [39]. 

The maximum root length of the wheat sapling was seen with the 150 ppm of ZnONPs, which was 11.68 cm, and the lowest was with the control sapling, which was just 6.32 cm. The maximum shoot length of wheat sapling was obtained when the ZnONPs concentration was 100 ppm, which reached up to 13.8 cm, and the lowest was in control, which was just 9.34 cm. The maximum fresh weight of wheat sapling root was obtained at 200 ppm, which was 0.158 gram, and the lowest was 0.04 gram in the control sapling. Here, the highest weight of the fresh weight of soot of wheat sapling was also obtained at 200 ppm of ZnONPs, while the lowest fresh weight of soot was obtained when the ZnONPs ppm was 0 i.e., with the control, which was 0.048 gram. Previously, Adrees et al. 2021, Tondey et al. 2021 also reported that Zn is a micronutrient, required by the plant, and promotes the growth of the wheat plant, but at a higher concentration of ZnONPs leads to phytotoxicity [42]. Consequently, there will be retarded growth of the plant in the form of short roots or soot length, their wet weight and dry weight [43,44].

## 4. Conclusions

There are several nanoparticles that could act as a nutrient for plants in small quantities. There is a possibility that the higher dose may have an adverse effect on the growth of the plant. The biosynthesized zinc oxide nanoparticles of size 20–80 nm was capped with several biomolecules present in the onion peel which helped in controlling the size of the nanoparticles. The zinc ion is an essential mineral for plants, which is required for activation of enzymes for performing various activities of plants, but their availability in the nanosized form may increase their uptake efficiency by the plant. The effect of various doses of zinc oxide nanoparticles on mung bean and wheat seeds exhibited that ZnONPs at lower to high concentrations enhances the seedling growth, seed germination percentage and fresh as well as dry weight, indicating that the concentration of ZnO up to 600 ppm is not harmful to plants. The biological route for the synthesis of zinc oxide nanoparticles not only minimizes the waste generated from onion, but rather it also helps in the formation of biocompatible and eco-friendly zinc oxide nanoparticles.

## Figures and Tables

**Figure 1 materials-15-02393-f001:**
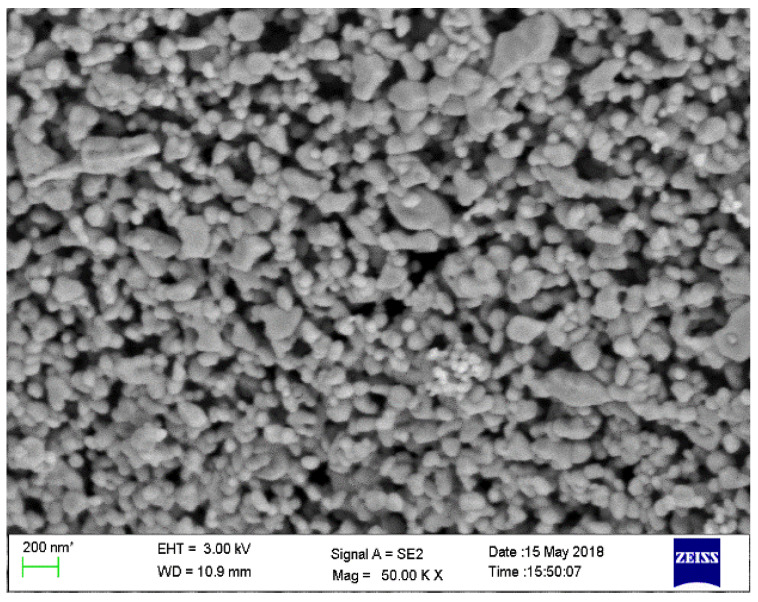
FESEM image of ZnONPs synthesized by onion peel extract at 200 nm.

**Figure 2 materials-15-02393-f002:**
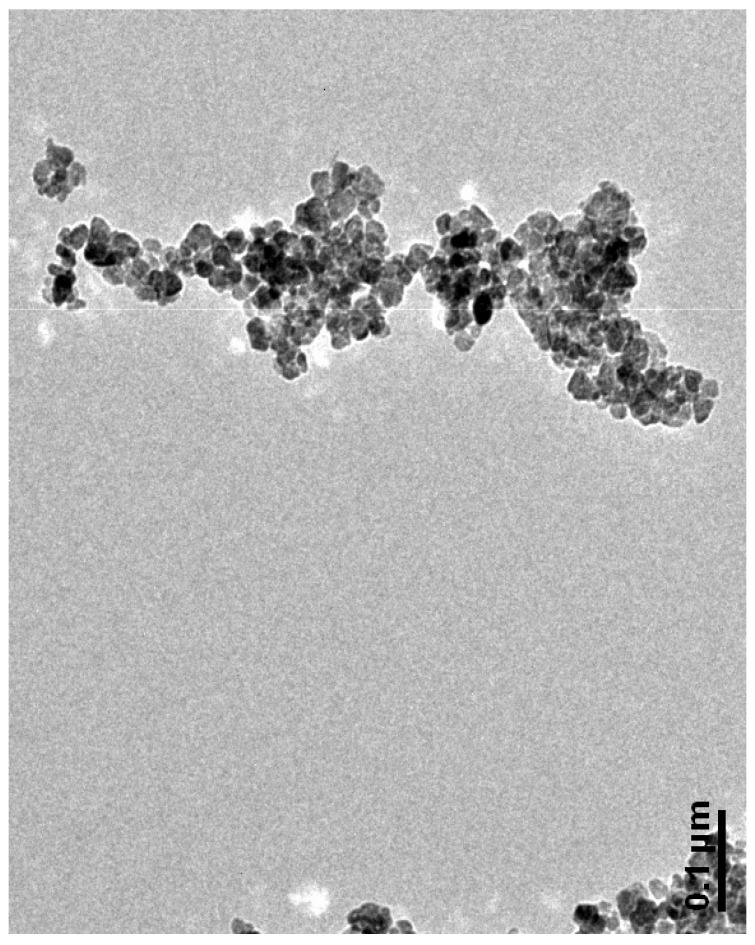
TEM image of onion peel mediated synthesized ZnONPs.

**Figure 3 materials-15-02393-f003:**
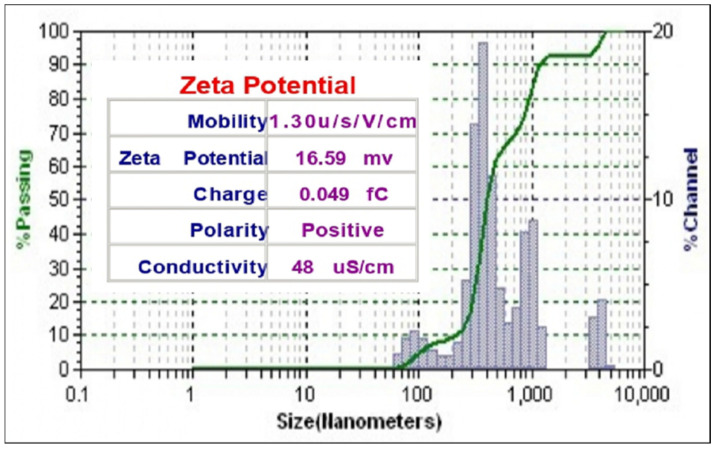
Particle size distribution of ZnONPs.

**Figure 4 materials-15-02393-f004:**
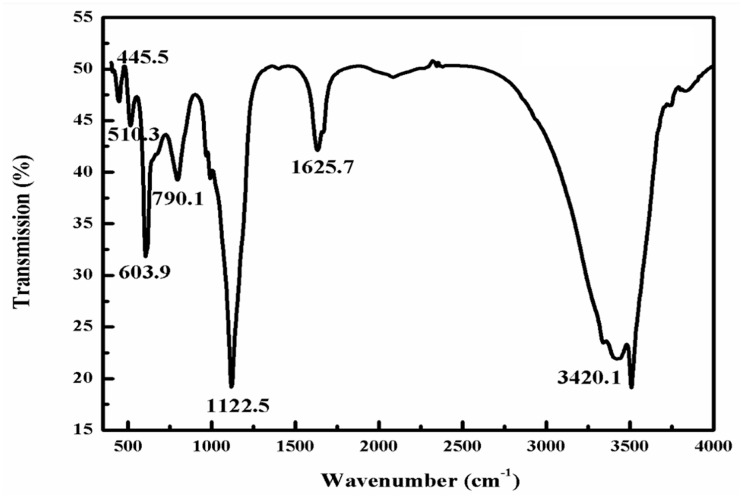
FTIR spectra of onion peel mediated synthesized ZnONPs.

**Table 1 materials-15-02393-t001:** Physico-chemical characterization of soil.

Sr.No	Parameter	Analysis Result	Quality
1	Total nitrogen	0.87%	high
2	Phosphorus (P_2_O_5_)Kg/Ac	5	low
3	Potassium (K_2_O) Kg/Ac	365	high
4	pH	7.60	normal
5	Conductivity	1.35	Salts are high
6	Sulfur (ppm)	16.5	Medium
7	Zn (ppm)	4.20	Medium
8	Fe (ppm)	8.40	Medium
9	Mn (ppm)	29.1	High
10	Cu (ppm)	1.92	High

**Table 2 materials-15-02393-t002:** Results for Mung bean seed.

Treatment	Germination Index	Seed Germination %	Seedling Growth (cm)		Fresh Weight (gms)		Dry Weight (gms)	
			Root Length	Soot Length	Root	Soot	Root	Soot
Control		90	8.066	9.34	0.052	1.182	0.025	0.114
50 ppm	79.42	100	9.14	11.45	0.108	1.638	0.042	0.198
100 ppm	76.49	100	9.49	12.8	0.066	1.215	0.024	0.122
150 ppm	96.67	100	7.51	13.38	0.106	1.734	0.035	0.188
200 ppm	101.95	100	7.12	15.88	0.056	1.596	0.021	0.150
400 ppm	104.45	100	6.95	13.44	0.081	1.341	0.034	0.168
600 ppm	105.67	100	6.87	12.42	0.091	1.625	0.029	0.215
800 ppm	119.67	90	6.74	12.22	0.078	1.513	0.031	0.156
1000 ppm	119.33	90	6.76	11.71	0.106	1.284	0.037	0.143
1200 ppm	110.30	80	6.50	10.54	0.060	1.260	0.019	0.131

**Table 3 materials-15-02393-t003:** Results for wheat seed.

Treatment	Seed Germination %	Seedling Growth (cm)		Fresh Weight		Dry Weight	
		Root Length	Soot Length	Root	Soot	Root	Soot
Control	50	6.32	9.34	0.040	0.212	0.0114	0.048
50 ppm	70	9.27	11.31	0.122	0.413	0.063	0.081
100 ppm	70	11.07	13.8	0.097	0.617	0.059	0.109
150 ppm	80	11.68	12.74	0.087	0.571	0.057	0.113
200 ppm	80	11.3	11.66	0.158	0.695	0.078	0.106
400 ppm	100	10.02	11.32	0.123	0.685	0.034	0.109
600 ppm	100	9.96	11.3	0.127	0.588	0.042	0.107
800 ppm	100	9.57	10.6	0.120	0.665	0.068	0.108
1000 ppm	80	8.38	10	0.087	0.442	0.045	0.083
1200 ppm	70	8.41	9.51	0.072	0.432	0.047	0.073

## Data Availability

All relevant data are included within the article.

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
