# Peer review of "Onion Peel Waste Mediated-Green Synthesis of Zinc Oxide Nanoparticles and Their Phytotoxicity on Mung Bean and Wheat Plant Growth"

_materials, 2022, doi:10.3390/ma15072393_

Round 1

Reviewer 1 Report

The authors synthesized zinc oxide nanoparticles (ZnONPs) using Allium cepa peel and evaluated the effect on the growth of Vigna radiate and Triticum aestivum. Work is acceptable for publications after the following concern is addressed.

The authors state:

“size revealed by the DLS is many fold larger than the microscopy because the size by DLS is in hydrodynamic size while the size revealed by microscopy is in dry form.”

I disagree with this statement. I think the reason why the average size is larger is related to DLS being more sensitive to aggregation. Of course, shape and hydration have an effect on particle size but I think this is marginal in their case. I see their histogram (Fig. 3) and I am wondering if DLS samples were filtered. I also suggest to include the corresponding correlation function which shows that the baseline value of 1 is achieved at long correlation times.

Author Response

The authors synthesized zinc oxide nanoparticles (ZnONPs) using Allium cepa peel and evaluated the effect on the growth of Vigna radiate and Triticum aestivum. Work is acceptable for publications after the following concern is addressed.

The authors state:

  1. “size revealed by the DLS is many fold larger than the microscopy because the size by DLS is in hydrodynamic size while the size revealed by microscopy is in dry form.”

I disagree with this statement. I think the reason why the average size is larger is related to DLS being more sensitive to aggregation. Of course, shape and hydration have an effect on particle size but I think this is marginal in their case. I see their histogram (Fig. 3) and I am wondering if DLS samples were filtered. I also suggest to include the corresponding correlation function which shows that the baseline value of 1 is achieved at long correlation times.

A/R: Thank you for this valuable comment and suggestion. Authors have now rectified the sentence as per the suggestion of the reviewer in the revised manuscript.

Reviewer 2 Report

This manuscript presents the effect of ZnO nanoparticles made from onion peels on the germination of mung bean and wheat seeds. The manuscript was organized and written well. Materials and Methods section was nicely written. Below are some minor comments for consideration.

It would be good if the manuscript is reviewed by an English language professional or editor to fix any English language errors. I have noted a few English errors below.

The authors should check the journal formatting guidelines to follow consistent formatting for the citations of the references given in the text. The authors have used reference numbers in square brackets, authors name and year format, year with or without parenthesis, etc.

Line 50:  remove commas in “,employs,”.

Line 54:  It would be good to provide the common name of Allium cepa as “onion”.

Line 56:  Please change “considered” to “considered to have”.

Line 60:  Please correct “reportedthe”.

Lines 68-69:  Pleas provide citations at the end of this line to indicate the few authors who made ZnO NPs from onion peel.

Line 84:  Please delete “was” from “was collected”.

Line 87:  Please provide the name and model of the grinder used for the study.

Line 91:  Please correct “In the next, step,”.

Line 140:  Please check “socked seeds”. Are they “soaked seeds”?

Line 154:  Pease check the formula for the “Relative seed germination Rate”. I think it should be “(Ss/Sc) x 100.”

Line 160:  Where are the results for “Seedling vigour index”? I could not find them in the manuscript.

Line 179:  In Figure 1, where is the part “(a) at 1 um”?. Also, are you using the symbol “u” to mean “micron”. If so, it would be good to use the correct symbol.

Line 197:  Please correct “small smaller”.

Line 233:  Please revise “was present” to “were present”.

Line 244:  Please correct “80&”.

Author Response

This manuscript presents the effect of ZnO nanoparticles made from onion peels on the germination of mung bean and wheat seeds. The manuscript was organized and written well. Materials and Methods section was nicely written. Below are some minor comments for consideration.

  1. It would be good if the manuscript is reviewed by the English language professional or editor to fix any English language errors. I have noted a few English errors below.

The authors should check the journal formatting guidelines to follow consistent formatting for the citations of the references given in the text. The authors have used reference numbers in square brackets, authors name and year format, year with or without parenthesis, etc.

A/R: Thank you for this valuable comment and suggestion. The authors have now thoroughly edited the manuscript as per the suggestion of the reviewer in the revised manuscript.

  1. Line 50:  remove commas in “,employs,”.

A/R: Thank you for this valuable comment and suggestion. The authors have now removed the commas in “, employs,” in the revised manuscript.

  1. Line 54:  It would be good to provide the common name of Allium cepaas “onion”.

A/R: Thank you for this valuable comment and suggestion. The authors have now provided common name i.e. onion for Allium cepa as in the revised manuscript as suggested by the reviewer in the revised version of the manuscript.

  1. Line 56:  Please change “considered” to “considered to have”.

A/R: Thank you for this valuable comment and suggestion. The authors have now changed the suggestion in the revised manuscript.

  1. Line 60:  Please correct “reportedthe”.

A/R: Thank you for this valuable comment and suggestion. The authors have now rectified the mistake in the revised manuscript as suggested by the reviewer.

  1. Lines 68-69:  Please provide citations at the end of this line to indicate the few authors who made ZnO NPs from onion peel.

A/R: Thank you for this valuable comment and suggestion. The authors have now provided citations at the end of the suggested line in the revised manuscript as suggested by the reviewer.

  1. Line 84:  Please delete “was” from “was collected”.

A/R: Thank you for this valuable comment and suggestion. The authors have now rectified the mistake in the revised version of the manuscript

  1. Line 87:  Please provide the name and model of the grinder used for the study.

A/R: Thank you for this valuable comment and suggestion. The authors have now provided details of the grinder used for the grinding of the onion peel in the revised manuscript as suggested by the reviewer.

  1. Line 91:  Please correct “In the next, step,”.

A/R: Thank you for this valuable comment and suggestion. The authors have now rectified the mistake in the revised version of the manuscript

  1. Line 140:  Please check “socked seeds”. Are they “soaked seeds”?

A/R: Thank you for pointing out this mistake. Yes the word was soaked seeds, which the authors have now corrected in the revised version of the manuscript

  1. Line 154:  Pease check the formula for the “Relative seed germination Rate”. I think it should be “(Ss/Sc) x 100.”

A/R: Thank you for this suggestion. No the formula is correct.

  1. Line 160:  Where are the results for “Seedling vigour index”? I could not find them in the manuscript.

A/R: Thank you for this valuable comment and suggestion. The authors now removed this sentence from the revised version of the manuscript.

  1. Line 179:  In Figure 1, where is the part “(a) at 1 um”?. Also, are you using the symbol “u” to mean “micron”. If so, it would be good to use the correct symbol.

A/R: Thank you for pointing out this mistake. The authors have now rectified the mistake in Fig.1 Authors have also changed the symbol of “u” in the revised manuscript as suggested by the reviewer.

  1. Line 197:  Please correct “small smaller”.

A/R: Thank you for pointing out this mistake. The authors have now rectified the mistake in the revised manuscript as suggested by the reviewer.

  1. Line 233:  Please revise “was present” to “were present”.

A/R: Thank you for pointing out this mistake. The authors have now rectified the mistake in the revised manuscript as suggested by the reviewer.

  1. Line 244:  Please correct “80&”.

A/R: Thank you for pointing out this mistake. The authors have now rectified the mistake in the revised manuscript as suggested by the reviewer.

Reviewer 3 Report

Minors: 

The scientific/colloquial names of the spp are not well defined and sometimes raises doubts about the plant in question. Consider to verify it all along the manuscript. 

Consider to correct the excess of spaces along the text. 

Consider to define the chemical acronym for all the compounds ZnONPs / ZnO NPs, Sulphur/ S, Zinc/Zn and Fe/Iron.

The references should be normalized as Numeric or Parenthetical (author-date). 

Consider to increase the Introduction section also the number of references.  Some references could not be found in the references section; please consider an extensive revision at this point.

The quality of figure 3 must be improved.

Author Response

  1. The scientific/colloquial names of the sppare not well defined and sometimes raises doubts about the plant in question. Consider to verify it all along the manuscript. 

A/R: Thank you for pointing out this mistake. The authors have now corrected the use of plant species name in the revised manuscript as suggested by the reviewer. 

  1. Consider to correct the excess of spaces along the text. 

A/R: Thank you for pointing out this mistake. The authors have now corrected the excess spaces along the text in the revised manuscript as suggested by the reviewer.

  1. Consider to define the chemical acronym for all the compounds ZnONPs / ZnO NPs, Sulphur/ S, Zinc/Zn and Fe/Iron.

A/R: Thank you for this valuable comment and suggestion. The authors have now defined the said chemical acronym for all the compounds in the revised manuscript as suggested by the reviewer.

  1. The references should be normalized as Numeric or Parenthetical (author-date). 

A/R: Thank you for this valuable comment and suggestion. The authors have now normalized the references as numeric throughout the manuscript as suggested by the reviewer in the revised manuscript.

  1. Consider to increase the Introduction section also the number of references.  Some references could not be found in the references section; please consider an extensive revision at this point.

A/R: Thank you for this valuable comment and suggestion. The authors have now increased the content in the introduction section as suggested by the reviewer in the revised manuscript. Besides this the authors have also provided the missing references in the reference section as suggested by the reviewer in the revised manuscript.

  1. The quality of figure 3 must be improved.

A/R: Thank you for this comment and suggestion. The authors have now improved the quality of Figure 3 in the revised manuscript as suggested by the reviewer.

Round 2

Reviewer 1 Report

Authors addressed DLS interpretation, which is the only issue mentioned in their response. The authors' response to reviewer does not mention any other change related to DLS method. I could not detect in their manuscript any comment on whether DLS samples were filtered (what size then?) and could not see any DLS time correlation functions (are correlation functions reaching their baseline?). We often get these types of distribution in the presence of dust, which makes the interpretation of correlation function in terms of diffusion modes (and related particle sizes) unreliable.  I recommend for publication after the authors report a more complete description of their DLS experiments.

Author Response

Comments and Suggestions for Authors

Authors addressed DLS interpretation, which is the only issue mentioned in their response. The authors' response to reviewer does not mention any other change related to DLS method. I could not detect in their manuscript any comment on whether DLS samples were filtered (what size then?) and could not see any DLS time correlation functions (are correlation functions reaching their baseline?). We often get these types of distribution in the presence of dust, which makes the interpretation of correlation function in terms of diffusion modes (and related particle sizes) unreliable.  I recommend for publication after the authors report a more complete description of their DLS experiments.

A: Thanks for the comment. The overall particle size appears larger due to the formation of the hydrodynamic layer as the DLS measurement is done in the liquid medium, where there are higher chances of aggregation due to the dispersion medium. But when we have done the TEM analysis we got the actual particle size of the synthesized nanomaterial.

We did not use any type of filter or syringe filter while analysing the DLS, so the authors are agreed that the sample may contain dust particles. Moreover, the author did not obtain the correlation graph by DLS but yes they have the same for particle size analyser.